# Interfacial Adhesion and Mechanical Properties of Wood-Polymer Hybrid Composites Prepared by Injection Molding

**DOI:** 10.3390/polym13172849

**Published:** 2021-08-25

**Authors:** Alexander Stadlmann, Andreas Mautner, Maximilian Pramreiter, Alexander Bismarck, Ulrich Müller

**Affiliations:** 1Department of Material Science and Process Engineering, Institute of Wood Technology and Renewable Materials, University of Natural Resources and Life Sciences Vienna, Austria (BOKU), Konrad Lorenz Strasse 24, 3430 Tulln an der Donau, Austria; maximilian.pramreiter@boku.ac.at (M.P.); ulrich.mueller@boku.ac.at (U.M.); 2Polymer and Composite Engineering (PaCE) Group, Faculty of Chemistry, Institute of Material Chemistry and Research, University of Vienna, Währinger Strasse 42, 1090 Vienna, Austria; andreas.mautner@univie.ac.at (A.M.); alexander.bismarck@univie.ac.at (A.B.); 3Department of Chemical Engineering, South Kensington Campus, Imperial College London, London SW7 2AZ, UK

**Keywords:** interfacial bond strength, wood-polymer composites, wood-polymer interface, XPS

## Abstract

Birch (*Betula pendula* Roth.) and beech (*Fagus sylvatica* L.) solid wood and plywood were overmolded with polyamide 6 (PA 6) and polypropylene (PP) to investigate their mechanical properties and interfacial adhesion. In the case of PA 6, maximum tensile shear strengths values of more than 8 to 9 MPa were obtained for birch and beech, respectively. The values are comparable to bond strengths of commercial joints bonded with formaldehyde-containing amino-plastics. Perpendicular to the wood elements, bond strength values of 3 MPa was achieved for PA 6. The penetration of the polymers into the wood structure results in a non-densified interphase and subsequent plastic deformation of the wood structure beyond the interphase. These compressed areas influenced the interfacial adhesion and mechanical interlocking. SEM and XPS analysis revealed different interpenetration behavior of the polymers into the wood structure, with chemical interaction confirmed only for wood and PA 6 but not PP.

## 1. Introduction

The mobility sector of the European Union is responsible for 26% of its total CO_2_ emissions. Around one fourth of these emissions are driven by the weight of the vehicle [1], increasing the importance of lightweight materials for automotive parts like wood [2]. Manufacturing technologies such as milling, cutting, gluing, molding, etc. for the production of wood-based products are well studied and established in the wood industry. In order to introduce wood and wood-based materials in these new areas of application such as the automotive industry, it is necessary to consider new production technologies during the design of wood-based hybrid components. Mair-Bauernfeind et al. [3] investigated the sustainability of wood and wood-based materials compared to other materials such as steel, where wood showed environmental, economic and social advantages. In addition, wood has also been increasingly used in multi-story buildings in the form of wood-concrete hybrid construction for several years. As claimed by Franzini et al. [4], the bio-based material wood also offers better indoor air quality, lower carbon dioxide emissions and competitive costs compared to concrete. Due to its sustainable nature and comparably low density, in recent years there has been an increasing demand for the utilization of wood and renewable materials in the mobility sector [5,6,7,8,9]. Besides utilizing sustainable and renewable products, formaldehyde-free bonding and joining of wood get more and more importance. The total amount of adhesive in plywood production can reach levels up to 20% for continuous bond lines that require high loading [10,11]. Kohl et al. [7] presented the environmental impact of urea-formaldehyde bonded beech plywood. Commonly, manufactured structural components in this field are made from steel, aluminum, magnesium, polymers or polymer composites by means of pressing, deep drawing, casting and molding [12]. Due to their low resistance against various media—e.g., salts—metals are usually coated using synthetic coating systems [13]. In contrast to metals, components made of polymers that are stabilized with suitable additives exhibit better durability and weathering behavior [14]. However, low mechanical properties and poor creep behavior of polymer-based components without fiber reinforcement (e.g., dashboards, claddings or wheel cases) negatively affect the applicability for load-bearing components in automotive parts. In addition to the properties of wood in terms of load-bearing capacity and lower density compared to most polymers, wood also provides good resistance to accelerated weathering in salty environment [15]. Therefore, back-injection molding or overmolding is a suitable technique for combining the good properties of wood with the durability, weather resistance and elasticity of polymers [16] to create wood-polymer hybrid composites. In addition, functional parts, such as brackets or mounting aids, can be easily fabricated by injection molding.

Wood as a bio-based reinforcement in polymers, i.e., in wood-polymer composites (WPCs), is well studied and widely used in applications such as furniture, decking, automotive and building components [17,18]. The mechanical interlocking and the mechanical adhesion as well as the effect of different wood species on the wood-polymer interaction of WPCs have already been investigated [19,20]. Gacitua et al. [19] observed that molten polymer (high density polyethylene, HD-PE) penetrates into the wood micro-structure resulting in a mechanical interaction between polymer and wood. Furthermore, the viscosity of the polymer melt also influences the penetration behavior [20]. Further research was carried out by Sretenovic et al. [21] to better understand the micro-mechanical behavior of wood plastic composites (WPC), demonstrating stress transfer from the wood to an LDPE plastic matrix caused by mechanical interlocking.

The modification of wood fibers for WPC production with various coupling agents aiming to improve interfacial adhesion, thus increasing strength and impact properties to a large extent, is well studied. Keener et al. [22] investigated the interaction of different coupling agents, i.e., maleic anhydride, polyolefins and peroxides in agrofiber polypropylene (PP) and polyethylene (PE) composites. In PE composites those coupling agents triple the impact bending strength and double the tensile strength, whereas the strength of PP composites increased by more than 60%. Correa et al. [23] aimed to improve the adhesion of wood-flour PP composites using maleated coupling agents and observed an increased interfacial adhesion between the matrix and fibers, which led to an improved load transfer and thus increased mechanical properties.

Polymers, used as adhesive in plywood fabrication have been investigated by Fang et al. [24] and Chang et al. [25] using HD-PE to bond poplar veneers by hot-pressing. The influence of the moisture content (MC) of the veneers, pressing temperature and pressure as well the quantity of the HDPE films on the mechanical and physical properties was investigated and compared with conventional urea-formaldehyde adhesive bonded plywood. They found that the MC of the veneers affected HDPE penetration. With increasing MC, the penetration depth of the polymer melt into the vessels (pores) of the wood structure decreased, which resulted in lower mechanical properties. Increased pressing temperature and pressure increased the bond strength as more polymer melt did penetrate into the vessels, thereby improving the mechanical properties. Furthermore, the dimensional stability could be improved when using HDPE of a higher quality. Surface modification using silane agents to improve the performance of wood-polymer plywood was also investigated by Fang et al. [26], Liu et al. [27] and Bekhta et al. [28], which results in a significant increasing tensile shear strength and a reduction of water uptake and lead to an improved dimension stability. Regarding particleboard production, there are several formaldehyde-free synthetic and renewable adhesive systems available, with the main drawback being availability and higher costs [29]. Overall, the mechanical performance of pure wood adhesives was also well discussed by Stoeckl et al. [30], where a wide range in stiffness was found. Briefly, the commonly used adhesives in engineered wood products and wood composites [29,30,31,32], as well as so called wood-plastic composites (i.e., extrusion and injection molding of wood fibers and particles with different kinds of plastics) [33], and improvements with various coupling agents is well investigated so far. However, almost no research was found on wood-polymer hybrid composites produced by means of injection molding. However, the realization of directly overmolded wood will help to reduce production time and costs, number of production steps, formaldehyde emission and carbon footprint. 

The present study aims to investigate mechanical properties and in particular the interfacial adhesion of wood-polymer hybrid composites prepared by injection molding. A frequently used wood-based material for non-structural automotive is plywood. Therefore, birch (*Betula pendula* Roth.) and beech (*Fagus sylvatica L.*) plywood boards were overmolded with PP at varying injection temperatures to investigate the effect of the injection temperature on the mechanical properties. A novel test setup was established to evaluate the tensile shear strength, the tensile strength perpendicular to the plane of the board and the tensile strength perpendicular to the edge. In addition to overmolded plywood, the tensile shear strength of birch and beech solid wood specimens overmolded with PP and polyamide 6 (PA 6) was investigated. To investigate the penetration depth of the polymer-melt into the wood micro-structure and the polymer wood adhesion, X-ray photoelectron spectroscopy (XPS) was performed. Furthermore, the interphase between the wood and the polymer was investigated by means of scanning electron microscopy (SEM). The main research questions of this study were as follows:

Q1. Does the polymer melt penetrate into the wood structure and what determines the adhesion between the polymer and wood, including chemical bonds?

Q2. Is the interfacial adhesion of an overmolded wood-polymer hybrid composites as strong as bonded wood products with commercial adhesives?

## 2. Materials and Methods

### 2.1. Materials

Industrially manufactured birch and beech plywood, with a thickness of 10 mm, and solid wood were sourced from Frischeis GmbH (Stockerau, Austria). Polypropylene (Daplen KSR 4525) and polyamide 6 (Grilon BZ 3) were provided by Borealis (Vienna, Austria) and EMS-Chemie AG (Domat, Switzerland), respectively.

### 2.2. Plywood Composites

Sixteen boards of each species, birch and beech, respectively, with a dimension of 297 × 146 mm were prepared using a circular saw and overmolded with PP as shown in Figure 1a. Injection molding was performed with an injection molding machine (Engel ES 1350/200 HL-V, Schwertberg, Austria) with a screw diameter of 70 mm. Each species was overmolded at three different injection temperatures to investigate the effect of the injection temperature on the interfacial adhesion and the mechanical properties of the wood-polymer composites. For this, cylinder temperatures were set to an average value of 220 °C, 240 °C and 260 °C, respectively. The volumetric flow rate was 15 cm^3^/s at an injection pressure of 170 bar, injection time of 2 s and the cycle time was 80 s. The way point of the feed screw was set to 35 mm, which corresponds to a changeover point at 95 cm^3^ after an injection time of 10 s and a holding pressure of 40 bar. 

In order to investigate mechanical properties under three different load conditions, specimens were cut using a circular saw (Figure 1a–d). For tensile tests perpendicular to the plane of the board (σ_P_), specimens with a dimension of 50 × 20 mm (Figure 1b), and for tensile tests perpendicular to the edge (σ_E_) specimens with a dimension of 120 × 20 × 10 mm were produced (Figure 1c), with one specimen consisting of 2 individual parts. These two parts were welded together with a welding mirror. For this purpose the temperature of the welding mirror was set to 200 °C and the welding time was 2 s. Specimens for tensile shear (σ_S_) tests, performed according to DIN EN 302-1, had a dimension of 120 × 20 × 4 mm, with the overlap length of the overmolded areas being 10 mm [34] (Figure 1d). In total 121 birch and 118 beech samples were prepared for this study.

### 2.3. Solid Wood Composites

Forty board specimens were cut with a dimension of 140 × 140 mm from each solid wood species, planed to a thickness of 4 mm and overmolded with PP or PA 6, respectively, to a final thickness of 8 mm (Figure 2a). Injection molding was performed with an injection molding machine (Wittmann Battendfeld Smart Power 120/750 B 8) having a screw diameter of 70 mm. 

For PA 6, the cylinder temperature was set to an average value of 260 °C. The volumetric flow rate was 40 cm^3^/s at an injection pressure of 360 bar, the injection time was 2 s and the cycle time was 64 s. The changeover point was set to 12 cm^3^, with a holding pressure and time of 100 bar and 2 s, respectively.

For PP, the cylinder temperature was set to an average value of 260 °C, the volumetric flow rate to 15 cm^3^/s, the injection pressure was 170 bar, the cycle time 66 s and injection time 5 s. The changeover point was set at 12 cm^3^, with a holding pressure and time of 100 bar and 1 s, respectively. In total 108 specimens with a dimension of 110 × 20 × 8 mm were produced using a circular saw (Figure 2b). In total 50 birch and 58 beech samples were prepared.

### 2.4. Mechanical Properties of Wood-Polymer Composites

Mechanical tests were performed using a universal testing machine (Zwick/Roell Z20, Ulm, Germany). Prior to mechanical tests, all samples were stored under standard climate conditions (20 °C ± 2 °C, 65% ± 5% relative humidity) according to standard ISO 554 [35] until constant mass was reached. All tests were stopped after a 30% load reduction of the maximum force (F_max_) was reached or failure occurred within 90 ± 30 s.

For plywood composites tensile tests perpendicular to the plane of the board (σ_P_) were performed with clamps originally designed for testing internal bond strength of particle and fiber boards, [36] which were used to attach the wooden part to the testing machine as shown in Figure 3a. A pre-force of 10 N was applied before testing at a constant crosshead speed of 1 mm/min. σ_P_ was calculated according to DIN 52 188 [37], by dividing F_max_ through the calculated interface area. σ_E_ was determined following DIN 52 188 [37], depicted in Figure 3b. After a pre-force of 10 N was applied, the specimens were loaded at a constant speed of 0.3 mm/min. σ_S_ was determined following DIN EN 302-1 [34] (Figure 3c) with an applied pre-force of 20 N at a constant speed of 0.4 mm/min.

For solid wood composites, the tensile shear strength (σ_S_) of birch and beech wood composites was assessed according to DIN EN 302-2 [34]. The samples were loaded with a pre-force of 10 N and tested at a constant crosshead speed of 0.6 mm/min.

### 2.5. Scanning Electron Microscopy (SEM) and X-ray Photoelectron Spectroscopy (XPS)

To investigate the penetration of the polymer into the wood structure on a microscopic level, two samples per combination were analyzed by means of SEM (Hitachi TM3030, Tokyo, Japan). To analyze the interphase of a cross-section of the overmolded samples, specimens with a dimension of about 3 × 8 mm were cut using a double-bladed circular saw. To obtain a smooth surface without any disturbing artefacts, the area of interest was cut with a razor blade.

XPS spectra were recorded to determine the penetration depth of the polymer into the microstructure of wood as well as chemical interactions between polymer and wood and to gain a deeper understanding into interfacial adhesion. Six solid wood specimens of each combination having a cross section of 8 × 4 mm were cut. The analysis was performed using an XPS system (Nexsa, Thermo-Scientific, Waltham, MA, USA) using an Al K_α_ radiation source operating at 72 W and an integrated flood gun. A pass energy of 200 eV, “Standard Lens Mode”, CAE Analyzer Mode and an energy step size of 1 eV for the survey spectrum were used. The diameter of the X-ray beam was 100 µm. A line scan was performed where four analysis points were placed in the wood-polymer interphase (Figure 4, −2 to +1). Starting from the first analysis point (Figure 4, 0), two spots were placed in the wood direction (Figure 4, −1 to −2) and one measuring spot in the polymer direction (Figure 4, +1) at a distance of 200 µm, respectively. As a reference, additional analysis points were placed in the wood substrate and in the polymer bulk, respectively. Prior to analysis the surface was cleaned by sputtering with Ar-clusters (1000 atoms, 6000 eV, 1 mm raster size) for 60 s. High-resolution spectra of C_1s_, N_1s_ and O_1s_ of 6 specimens were examined, acquired with 50 passes at a pass energy of 50 eV and an energy step size of 0.1 eV. These were analyzed using software package Thermo Avantage (v5.9914, Build 06617) with Smart background and Simplex Fitting algorithm by using Gauss-Lorentz Product. Peak profiles of C_1s_ and O_1s_ were deconvoluted.

### 2.6. Statistics

In this study a one-way analysis of variance with an error level of 0.05 was calculated using Excel 2016 (Microsoft, Redmond, WA, USA) to statistically evaluate the effect of the injection temperature and the wood species on the mechanical properties.

## 3. Results and Discussion

### 3.1. Mechanical Properties of Wood-Polymer Composites

#### 3.1.1. Plywood-Polymer Composites

Figure 5 displays the results of all test configurations at the different injection temperatures for birch and beech plywood, respectively, overmolded with PP. Regarding the influence of the injection temperature on the mechanical properties of these wood-polymer composites no statistically significant effect was present, with the exception of σ_E_ for the overmolded birch plywood specimens, for which higher injection temperatures resulted in higher σ_E_. Chang et al. [25] reported that the hot-pressing temperature and pressure exhibit an inflexion point, i.e., a certain pressure and temperature, at which penetration of the polymer into the wood structure with partly damaged cells and cracks and into the vessels is highest, thus resulting in the highest strength. Furthermore, no significant difference between birch and beech plywood for all test configurations was observed.

σ_P_ was on average (over all three injection temperatures) 3.16 ± 0.91 MPa and 2.89 ± 0.68 MPa for birch and for beech plywood, respectively. The highest σ_P_ for birch and beech plywood specimens was observed at an injection temperature of 240 °C (3.31 ± 1.15 MPa and 3.02 ± 0.49 MPa), similar to results reported by Chang et al. [25]. The tensile strength perpendicular to the edge was on average 5.08 ± 1.44 MPa and 4.78 ± 1.01 MPa for birch and beech plywood, respectively, with the highest values achieved at 260 °C for birch and 240 °C for beech plywood specimens (6.04 ± 1.04 MPa and 5.14 ± 1.24 MPa). Liu et al. [38] investigated the surface bond strength of engineered plywood in a similar fashion. Poplar veneers (*Populus tomentosa* Carriére) were bonded with chlorinated PP films on a wood fiber PP composite core layer (80% wood fiber and 20% PP) using a hot-pressing procedure. They observed surface bonding strength values of the veneers on the composite core layer, which is comparable with σ_P_, of approx. 1.75 MPa, which was significantly lower compared to our study. The higher values were attributed to two reasons. On the one hand, the poplar veneers used have a significantly lower tensile strength perpendicular to the grain of about 1.7–2.8 MPa compared to birch (~7.0 MPa) and beech (~7.0–10.7 MPa) wood [39]. On the other hand, they prepared the PP-bonded plywood using a hot-pressing process, in which the pressure applied was about 5 MPa at a temperature of 110 °C, which is much lower compared to those used in our study. Improved penetration of PP into the wood at 170 bar (17 MPa) and 360 bar (36 MPa) for PA 6, respectively, the pressures used in this study, could eradicate the damage caused in the wood structure due to compressive failure of the top layers.

σ_S_ of birch and beech plywood samples were on average 3.81 ± 0.76 MPa and 3.83 ± 0.69 MPa, respectively, with highest values observed at an injection temperature of 260 °C for birch and 220 °C for beech of 3.87 ± 0.68 MPa and 3.92 ± 0.94 MPa, respectively. Bekhta et al. [28] reported shear strength values for birch and beech plywood bonded with PA 6 and polyethylene (PE) higher than 3 MPa and 1.7 MPa, respectively. However, PP overmolded birch and beech plywood showed higher σ_S_, which is thought to be mainly influenced by the different process parameters used in this study.

#### 3.1.2. Solid Wood-Polymer Composites

Figure 6 summarizes the mean values, standard deviation and sample number for birch and beech solid wood, overmolded with PA 6 and PP, respectively. There was no statistically significant difference between the birch and beech solid wood composites. σ_S_ for birch-PA 6 were on average 5.71 ± 1.13 MPa, while σ_S_ for beech-PA 6 was slightly higher (6.36 ± 1.47 MPa). For PP-composites a slightly lower tensile shear strength was observed for birch solid wood compared to beech (2.33 ± 0.44 MPa and 2.54 ± 0.83 MPa, respectively). The observed maximum values for σ_S_ for the PA 6-composites were 8.65 MPa and 9.74 MPa and for PP-composites 2.98 MPa and 4.19 MPa for birch and beech, respectively. Compared to literature, the measured maximum values were similar, with tensile shear strengths of 9 MPa and 3.5 MPa reported for beech wood rods overmolded with PA 6 and PP, respectively [16]. However, a perfectly aligned longitudinal fiber orientation of the specimens results in fewer cut vessels and fibers and thus fewer open lumens into which polymer could penetrate, which in turn results in less mechanical interlocking and thus in a lower average tensile shear strength. For comparison, typical values for bonded birch and beech wood specimens (melamine-urea-formaldehyde (MUF), polyurethane (PU) and phenol-resorcinol-formaldehyde (PRF)) with commercially adhesives do exceed 10 to 11 MPa [31,32].

The differences between PA 6 and PP composites can be explained by the different polarity of both polymers. It is supposed that the polar PA 6 [40] exhibits a good adhesion and/or sound bonding with the wood surface, which promotes higher strength values. In addition, the high temperature during the molding process (260 °C) degrades free hydrophilic groups of wood polymers, mainly the hemicelluloses [41,42]. The effects of thermally modified wood fibers on the adhesion to thermoplastics were also reported by Follrich et al. [43]. It can be assumed, that exposure to elevated temperatures leads to a more hydrophobic character of the wood surface enhancing the interfacial compatibility to hydrophobic polymers, which results in improved interfacial interactions. Furthermore, the higher strength for PA 6-composites in contrast to PP-composites can also be explained by the higher cohesive strength of PA 6.

### 3.2. Wood-Polymer Interfaces

#### 3.2.1. Morphology of Wood Polymer Composites by SEM

Figure 7 shows cross sections of representative birch and beech solid wood composites. During the overmolding process, the melted polymers penetrated into the wood structure through the sliced vessels and fibers. For the specimens overmolded with PA 6, in comparison to PP-composites only minor penetration of the melt into the wood substrate was observed. Due to the high pressure used (360 bar) during the injection process, the melt flow was mainly directed in radial direction (Figure 7(1a–2b)). PP composites were fabricated at much lower injection pressure (170 bar), hence, the melt penetrated the wood cells in both directions, for birch and beech, respectively (Figure 7(3a–4b)). Furthermore, it was observed that the outer cellular structure (approximately 100 µm up to 200 µm) is stabilized by the polymer that penetrated the wood by filling the lumens of vessels and tracheids. Additionally, only a few micro gaps along the interface between the wood and the polymer were observed, which is interpreted as an indicator of good adhesion between the materials. In addition, wood rays and also the transition zone from early to late wood have a structurally reinforcing effect. As Mattheck and Kubler [44] presented, the many rays oriented perpendicular to the grain behave like beams, that lead to an increasing compressive strength of the wood structure. These compressed areas generate an increased interface and thus improved mechanical interlocking between the polymer and the wood surface. According to Sretenovic et al. [21] the mechanical interlocking influences the stress transfer from the polymer to the wood structure in wood fiber composites. In addition, Smith et al. [45] reported that both the porosity of the wood structure and the processing parameters are influencing mechanical interlocking. Furthermore, Gupta et al. [46] showed that there are strong correlations between surface roughness and interfacial adhesion.

Beyond the stabilized interface the wood structure exhibits a zone of compressive failure, caused by plastic deformation of the wood structure during injection molding (Figure 7(1a–4a)). These compressed zones range approximately 0.4 to 1.1 mm into the wood structure depending on the species. Müller et al. [47] reported that a pressure of about 12 MPa is required to densify a diffuse-porous wood structure (e.g., birch and beech) perpendicular to the grain. Corresponding to the thickness of the compressed zone, the overmolding procedure causes almost homogeneous densification across the overmolded surface for both birch and beech. However, the compression zone of birch wood is much larger than the compression zone of beech wood, which is caused by its lower compression strength perpendicular to the grain [39,48,49]. A lower ratio of strength perpendicular to the grain to the injection pressure, leads to higher densification of the wood substrate, causing the formation of a so-called weak boundary layer, which influences the strength of wood polymer composites. In birch wood samples, failure occurs mainly in the weak boundary layer, which corresponds to the results of the mechanical tests, both for solid wood and plywood overmolded with PA 6 and PP as well as findings, as reported by Chang et al. [25]. Furthermore, it is clearly shown, that mainly vessels are compressed in this range. In contrast, tracheids and wood fibers are compressed mainly in the peripheral areas up to a depth of 200 to 600 µm. 

#### 3.2.2. Elemental Composition and Chemistry of the Interface in Wood-Polymer Composites

From XP spectra information regarding the penetration of polymer into the wood structure and their interaction can be derived. The elemental composition (C, N and O) at various positions within the wood polymer interphase extracted from scans is shown in Table 1. Additionally, as an initial indicator of the presence of polymer in the wood structure and vice versa, the atomic ratio O/C and N/C for PA 6 composites and the O/C ratio for PP composites were calculated. For all composites, the O/C ratio significantly decreased from the wood substrate through the interphase towards the bulk polymer. Specifically, the birch-PA 6composites exhibited a constant decrease of the O/C ratio from the wood towards the polymer, whereas for beech wood samples this ratio significantly decreases in the interphase (Table 1, −1 to +1), after that the O/C ratio remained constant. Correspondingly, the N/C ratio significantly increased in the interphase to polymer direction (Table 1, −1 to +1). PP composites exhibited a similar trend regarding the O/C ratio; it significantly decreased from the wood substrate towards the interphase, both for birch and beech composites (Table 1, −1 to +1). These results confirm that penetration of the polymer melt into the wood cell wall structure takes place and not only through the cut vessels and fibers as determined in the SEM analyses. 

Carbon is the dominant element in both wood and polymer. For this reason, the carbon peak from high resolution spectra was deconvoluted into four components. With regard to wood, the C 1 peak (C-C or C-H) at approx. 284 eV corresponds to carbon-carbon or carbon-hydrogen bonds and is predominant in lignin or polymers such as PP. The C 2 peak at approx. 286 eV corresponds to carbon-non-carbonyl oxygen bonds (C-O), a major moiety in cellulose. The C 3 peak at approx. 287 eV is assigned to carbon atoms bound to two non-carbonyl oxygens (O-C-O) or to one carbonyl oxygen (C=O), while the C 4 peak at approx. 289 eV represents carboxylic groups (O-C=O) [50,51]. For PA 6 the C peak was deconvoluted into three components according to the literature [52], with the C 1 peak at approx. 284 eV corresponding to the aliphatic carbon atoms CH_2_ (**C-C** (C=O)-N-C), the C 2 at approx. 286 eV representing the carbon atoms linked to the amide nitrogen (C-C (C=O)-N-C) and the C 3 at approx. 287 eV representing the amide carbonyl group (C-C (**C=O)-N-**C). For aliphatic PP the C peaks were deconvoluted into two main components: C 1 peak (C-H or C-C) at approximately 284.5 eV and the C 2 peak (C-O) at approx. 286 eV [53]. To obtain information about the penetration as well the distribution of the chemical components, C 1 and C 3 peaks for specimens overmolded PA 6 and C 1 and C 2 peaks for specimens overmolded with PP were selected (Figure 8a–d and Appendix A).

In general, an increase of the measured atomic percentage of the C 1 components from the interface in polymer direction and at the same time a decrease in wood direction for all samples was found. For birch-PA 6 composites (Figure 8a), the amount of the C 1 component decreased significantly from the polymer bulk towards the interphase from 61.0 at.% to 51.0 at.% (Figure 8a, PA to 0), while the amount of C 1 remained almost constant through the interphase but decreased towards the wood substrate. For the C 3 component, the highest value of 27.5 at.% was observed at the interphase (Figure 8a, 0), significantly decreasing in both directions. For beech-PA 6 (Figure 8b) almost similar results were observed. The amount of the C 1 component decreased from the PA 6 bulk to the wood substrate from 51.0 at.% to 24.4 at.%, with the interphase area having a constant value. The highest amount of C 3 was again observed in the interphase of 33.9 at.% (Figure 8b, 0), decreasing toward the wood and polymer direction. The results of C 1 and C 3 indicate that the polymer melt penetrated the wood structure during injection molding up to about ~400 µm (Figure 8a,b, −2). Additionally, a nonlinear trend of ratios between C 1 and C 3 was observed indicating that chemical reactions, e.g., transamidation occurs influencing the ratio beyond the trend expected from pure mixing of polymer and wood [54].

Figure 8c,d shows the elemental distribution across the cross section of PP composites. Similar results were observed for birch and beech wood. Highest values of C 1 were determined in the polymer bulk, as expected (71.1 at.% and 68.4 at.% for birch and beech specimens, respectively). These amounts then constantly decreased towards to point 0 within the interphase of 44.9 at.% and 48.2 at.% (Figure 8c,d, 0) for birch and beech, respectively. Beyond point 0 a significant decrease of the C 1 component was observed towards wood (Figure 8c,d, −1). Between point −1 and the wood substrate no significant difference between C 1 for birch-PP and beech-PP can be observed. For C 2, the elemental composition also does not differ significantly from each other. PP does not interact with the wood structure beyond van der Waals interactions as PA 6 does, based on polar groups present in PA 6 and being absent in PP. Furthermore, PP only penetrated the wood structure up until 200 µm (Figure 8c,d, −1).

Results of the XPS analyses assist in explaining the results of the mechanical tests. Higher strength and stiffness of wood-PA 6 composites can be explained by PA 6 penetrating the wood structure on a macro- but also microscopic level, which corresponds to previous findings [16] where beech wood rods were overmolded with different polymer materials. In addition, chemical interaction of PA 6 with wood takes place due to the polar nature of the material, resulting in better interfacial adhesion as compared to PP and thus improved mechanical properties of wood-PA 6 composites produced by injection molding. However, sound bonding to the wood occurs, both for PA 6 as well as for the more hydrophobic material PP due to formation of an interphase by polymer penetration into the wood.

Based on the presented results, the initially proposed research questions can be answered as follows: The used polymers penetrate into the peripheral porous structure through the sliced vessels and fibers thus forming an interphase which contributes to adhesion by mechanical interlocking. In case of PA 6 wood composites, additional chemical interactions do seem to contribute to improved adhesion. Additionally, the mechanical properties of the produced (unmodified) wood-polymer composites can compete with commercially bonded wood-wood composites and, therefore, this technology is suitable to manufacture wood polymer hybrid composites for structural applications for instance for the automotive sector.

## 4. Conclusions

Solid wood and plywood were overmolded with PP and PA 6 in order to investigate the influence of process parameters on interfacial adhesion between wood and polymer and the mechanical properties of wood-polymer composites. The mechanical properties of these composites are influenced only to a small extent by the processing temperature used. Temperature effects on the wood substrate are of minor importance. However, SEM and XPS analysis showed that substantial amounts of molten polymer penetrated into the wood substrate. A weak boundary layer of compacted cells formed in the wood substrate, extending from the wood-polymer interphase to a depth of 1 mm. Due to the high-pressure during injection molding, a weak boundary layer consisting of heavy densified cells formed in the wood substrate, extending from the wood-polymer interphase to a depth of 1 mm. The weak boundary layer, which varied between birch and beech wood, lowered the mechanical properties of the wood-polymer composites. However, polymer interpenetrated the wood substrate through vessels, which led to the formation of a stabilized interphase and improved mechanical properties. Due to its polar character, PA 6 interacts chemically with the wood substrate, resulting in the highest tensile shear strength observed, ranging from 8 MPa to 9 MPa for birch and beech, respectively. Nevertheless, sufficient bonding and mechanical interlocking of PP was also observed for both wood species.

## Figures and Tables

**Figure 1 polymers-13-02849-f001:**
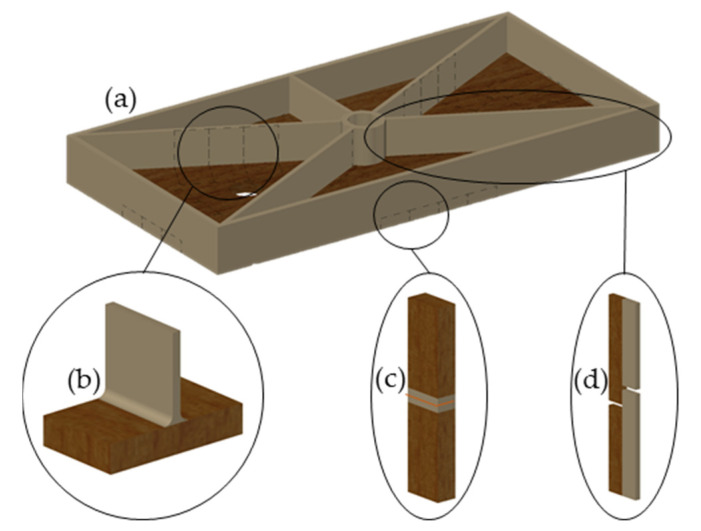
Schematic of the overmolded plywood boards (**a**) and schematic representation of the specimens for the tensile strength perpendicular to the plane of the board σ_P_ (**b**), tensile strength perpendicular to the edge σ_E_ (**c**) and tensile shear strength σ_S_ (**d**).

**Figure 2 polymers-13-02849-f002:**
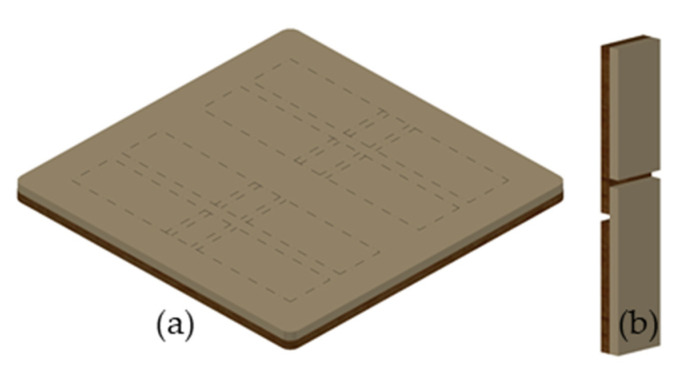
Schematic of the overmolded solid wood (**a**) and representation of the specimens for the tensile shear strength σ_S_ (**b**) according to ÖNORM EN 302-1 [34].

**Figure 3 polymers-13-02849-f003:**
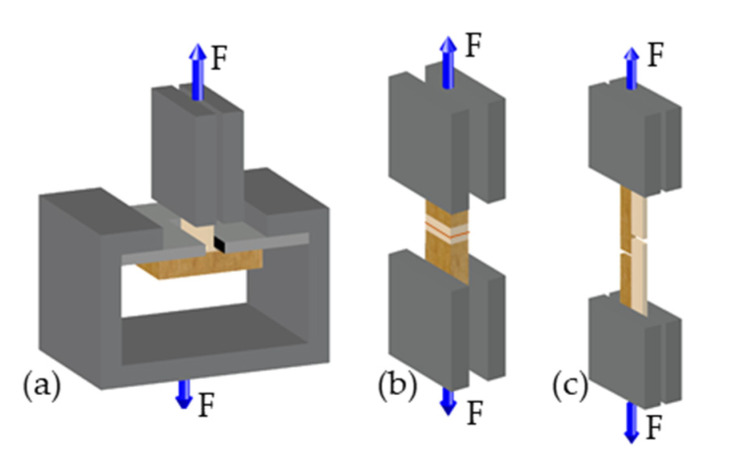
Schematic of the test setup for overmolded plywood specimens for strength measurements according to DIN 52 188 [37] and DIN EN 302-1 [34]: (**a**) test setup for tensile strength perpendicular to the plane of the board σ_P_, (**b**) test setup for tensile strength perpendicular to the edge σ_E_, and (**c**) test setup for tensile shear strength σ_S_.

**Figure 4 polymers-13-02849-f004:**
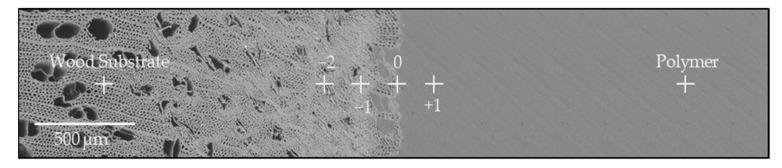
Micrograph of the overmolded solid wood specimens for X-ray photoelectron spectroscopy analyses (XPS) used to determine the elementary distribution of C, O and N and for high-resolution deconvoluted XPS spectra within the samples cross section, indicating the position of the X-ray beam.

**Figure 5 polymers-13-02849-f005:**
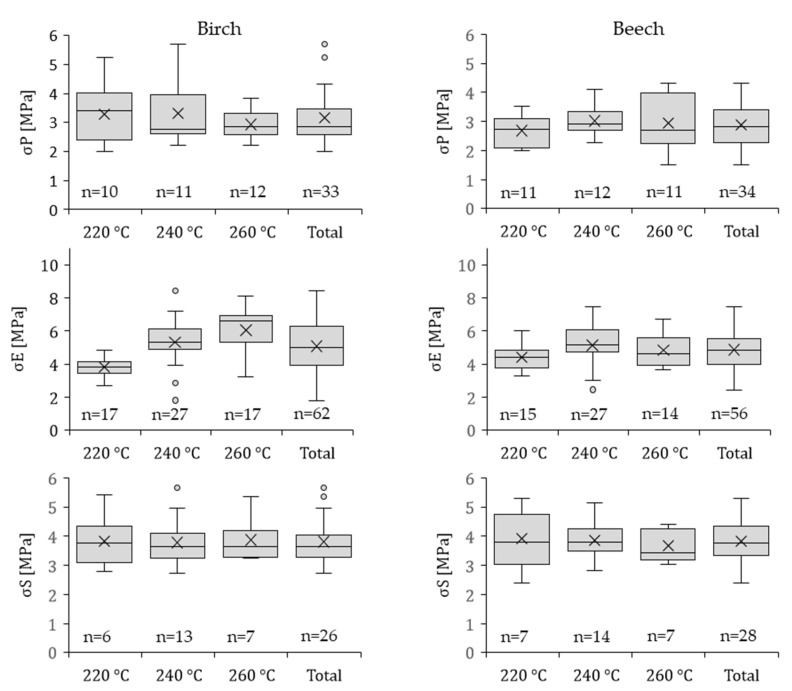
Strength properties of 121 birch plywood specimens and 118 beech plywood specimens overmolded with PP at three different injection temperatures. σ_P_ is the average tensile strength perpendicular to the plane of the board, σ_E_ is the average tensile strength perpendicular to the edge, σ_S_ is the average tensile shear strength and n is the number of the samples tested. The whiskers show minimum and maximum values. X is the mean value. ° indicate values of statistical outliers and—is the median.

**Figure 6 polymers-13-02849-f006:**
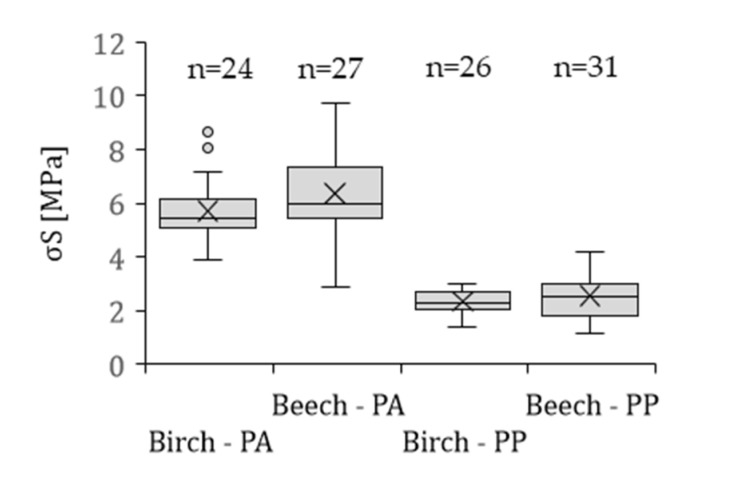
Average tensile shear strength σ_S_ of birch solid wood specimens and beech solid wood overmolded with PA 6 and PP, respectively; n is the number of the samples. The whiskers show minimum and maximum values. X is the mean value. ° indicate values of statistical outliers and—is the median.

**Figure 7 polymers-13-02849-f007:**
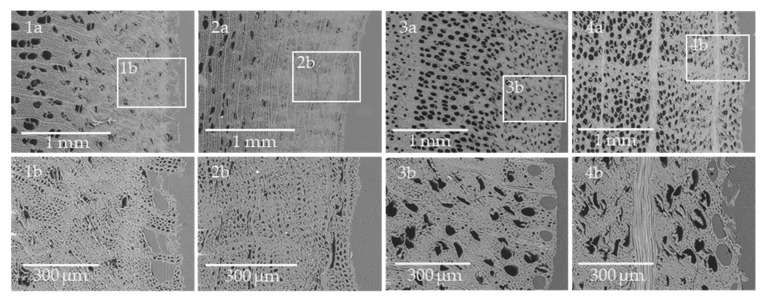
Representative SEM micrographs recorded on the cross-section of the overmolded specimens for birch solid wood (**1a**–**2b**) and beech solid wood (**3a**–**4b**); b represents always the detailed inset of a. (**1a**), (**1b**), (**3a**) and (**3b**) show samples overmolded in radial direction and (**2a**), (**2b**), (**4a**) and (**4b**) represents samples overmolded in tangential direction.

**Figure 8 polymers-13-02849-f008:**
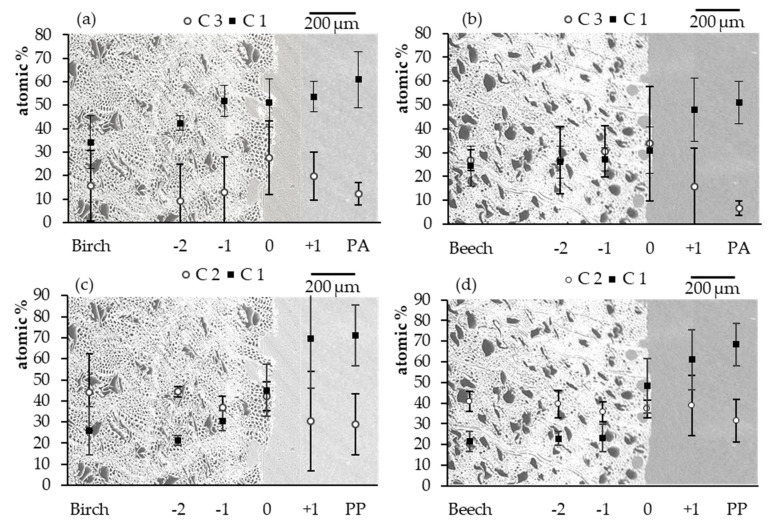
Mean values and standard deviation of the XPS results: Carbon peak components at C 1 and C 3 for birch solid wood (**a**) and beech solid wood (**b**) overmolded with PA 6 and the C 1 and C 2 for birch solid wood (**c**) and beech solid wood (**d**) overmolded with PP.

**Table 1 polymers-13-02849-t001:** Elemental composition of C, N and O over the sample cross-section for birch and beech solid wood overmolded with PA 6 and PP from XPS analysis. The position of the measuring spots placed on the samples are shown in Figure 4. In addition, the atomic O/C and N/C ratios determined by XPS analysis are presented.

Composite	Elemental	Wood Substrate	−2	−1	0	1	Polymer
Birch-PA 6	C at [%]	81.85	86.37	87.65	87.74	91.11	90.00
	N at [%]	0.19	0.14	0.15	2.05	4.28	5.02
	O at [%]	17.95	13.50	12.21	10.22	4.61	4.98
	O/C	0.219	0.156	0.139	0.116	0.051	0.055
	N/C	0.002	0.002	0.002	0.023	0.047	0.056
Beech-PA 6	C at [%]	67.77	67.76	67.37	73.07	89.24	87.74
	N at [%]	0.38	0.34	0.29	1.52	5.82	5.56
	O at [%]	31.85	31.90	32.34	25.41	4.95	6.71
	O/C	0.470	0.471	0.480	0.348	0.055	0.076
	N/C	0.006	0.005	0.004	0.021	0.065	0.063
Birch-PP	C at [%]	72.61	71.46	70.40	85.30	98.68	98.32
	N at [%]	0.45	0.27	0.33	0.35	0.41	0.40
	O at [%]	26.95	28.27	29.27	14.35	0.92	1.29
	O/C	0.371	0.396	0.416	0.168	0.009	0.013
Beech-PP	C at [%]	66.66	65.70	65.96	83.45	99.29	99.11
	N at [%]	0.32	0.28	0.24	0.38	0.30	0.31
	O at [%]	33.02	34.03	33.81	16.17	0.42	0.59
	O/C	0.495	0.518	0.513	0.194	0.004	0.006

## Data Availability

The raw/processed data required to reproduce these findings cannot be shared at this time due to legal or ethical reasons.

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
