# Peer review of "Interfacial Adhesion and Mechanical Properties of Wood-Polymer Hybrid Composites Prepared by Injection Molding"

_polymers, 2021, doi:10.3390/polym13172849_

Round 1
Reviewer 1 Report
- 1. overall, the science seems good.
- Does the scientific name need to be italicized? Check the standards. Maybe this is ok and is the journal standard.
- Introduction, you say “In In order….” say In order...
- you say, “The total amount of adhesive in plywood production can reach levels up to 20 %.” Perhaps add that this is a continuous bondline which requires such high loadings (for the non wood science reader).
- you say, “In contrast to metals, components made of polymers exhibit much better durability and weathering behavior,” this implies that all polymers are durable and weather resistant. Is this true? I don’ think you mean it this way, but that is how it reads.
- The introduction lacks a leading discussion about the automotive industry and instead the first sentence jumps straight into the automotive industry. So as a reader with no prior knowledge, I get confused.
- it says “Mair-Bauernfeind et al. investigated the sustainability of wood and wood-based materials compared to other materials such as steel, where wood showed environmental, economic and social advantages.” Since steel and concrete usually go together in buildings, you should also add to the manuscript that wood offers lower CO2 emissions, better indoor air quality, and competitive costs than concrete. I obtained this information from Franzini et al. in the forest products journal in volume seventy one, issue one and page sixty five. In our field, there is a big push in tall buildings / mass timber.
- It says, “ The mechanical interlocking and the mechanical adhesion as well as the effect of different wood species on the wood-polymer interaction has already been investigated.” If it has already been investigated, then why are you studying 2 different species in this study? You might want to clarify here to not conflict with your study.
- It says, “They found that the MC of the veneers is affecting the penetration of HDPE.” Say veneers affected HDPE penetration.
- it says, “Q2. Is the interfacial adhesion of an overmolded wood-polymer hybrid composites 113 as strong as bonded wood products with commercial adhesives?” But your introduction does not discuss the major adhesives used in plywood or wood composites. You need to add discussion and references around this from wood/forest products type journals. Try to use new journals if possible as many of the newer adhesives are hybrid mixtures.
- it says, “16 boards of each species, “ you should never start a sentence with a number. It should say Sixteen boards of each species….
- what was the rotation speed of the feed screw?
- what was the loading rate of the specimens? This is important since wood strength depends on the load rate.
- it says, “40 board specimens were,” same as earlier, do not start a sentence with numeric but instead spell out forty.
- Figure 3 seems like a nice diagram.
- Figure 5 – how come the sample sizes are not equal across different temperatures for any of the graphs? I would expect to see this once or twice with an explanation, but not all over the place.
16.it says, “ In addition, the high temperature during the molding process (260 °C) degrades free hydrophilic groups of wood polymers, mainly the hemicelluloses” you will degrade more than hemicellulose at that temperature. What about lignin or amorphous cellulose?
- it says, “In addition, wood rays and also the transition zone from early to late wood act a 322 reinforcement of the wood structure.” Do you need to discuss that rays are perpendicular to the earlywood/latewood rings and what effect this has on reinforcement?
- it says, “Müller et al. [41] reported that a pressure of about 334 12 MPa is required to densify a diffuse-porous wood structure.” Here you use pores instead of vessels to describe wood anatomy. When you first use the word vessel, earlier in the document, you could put pores in parenthesis.
- how does the C,N,O amount describe the cause of the chemical interface between wood and polymer?
Reviewer 2 Report
The paper deals with laminates investigation - wood/polymer. PP/PA matrices and birch wood are chosen for investigation. The Paper has sound, is well prepared, and discussed. There are no significant shortcomings in the paper. It can be published after a minor comment.
The reviewer suggesting the authors could discuss the question about developed bonding characteristics (chemical, physical, which bonds) in the investigated interphase between the birch substrate and polymer type and how it can be controlled and further enhanced.
